# A Hybrid Algorithm for Noise Suppression of MEMS Accelerometer Based on the Improved VMD and TFPF

**DOI:** 10.3390/mi13060891

**Published:** 2022-05-31

**Authors:** Yongjun Zhou, Huiliang Cao, Tao Guo

**Affiliations:** 1Science and Technology on Near-Surface Detection Laboratory, Wuxi 214000, China; 230109620@seu.edu.cn; 2Key Laboratory of Instrumentation Science & Dynamic Measurement, Ministry of Education, North University of China, Taiyuan 030051, China; guotao6@nuc.edu.cn

**Keywords:** High-G MEMS accelerometer (HGMA), multi-objective particle swarm optimization (MOPSO), variational modal decomposition, time-frequency peak filtering, denoising

## Abstract

High-G MEMS accelerometer (HGMA) is a new type of sensor; it has been widely used in high precision measurement and control fields. Inevitably, the accelerometer output signal contains random noise caused by the accelerometer itself, the hardware circuit and other aspects. In order to denoise the HGMA’s output signal to improve the measurement accuracy, the improved VMD and TFPF hybrid denoising algorithm is proposed, which combines variational modal decomposition (VMD) and time-frequency peak filtering (TFPF). Firstly, VMD was optimized by the multi-objective particle swarm optimization (MOPSO), then the best decomposition parameters [*k_best_*, *a_best_*] could be obtained, in which the permutation entropy (PE) and fuzzy entropy (FE) were selected for MOPSO as fitness functions. Secondly, the accelerometer voltage output signals were decomposed by the improved VMD, then some intrinsic mode functions (IMFs) were achieved. Thirdly, sample entropy (SE) was introduced to classify those IMFs into information-dominated IMFs or noise-dominated IMFs. Then, the short-window TFPF was selected for denoising information-dominated IMFs, while the long-window TFPF was selected for denoising noise-dominated IMFs, which can make denoising more targeted. After reconstruction, we obtained the accelerometer denoising signal. The denoising results of different denoising algorithms in the time and frequency domains were compared, and SNR and RMSE were taken as denoising indicators. The improved VMD and TFPF denoising method has a smaller signal distortion and stronger denoising ability, so it can be adopted to denoise the output signal of the High-G MEMS accelerometer to improve its accuracy.

## 1. Introduction

Thanks to the rapid development of micromechanical systems (MEMS) technology, the development and application of inertial sensor components have attracted extensive attention. As an outstanding representative of inertial sensors, the HGMA has been used in consumer electronics, aerospace and other high-precision measurement and control fields owing to its advantages of low cost and power consumption, high efficiency and sensitivity [1,2,3,4]. Due to the inherent defects of the hardware circuit and sensor itself, there is rich noise in the output signals of the accelerometer, which causes a large error and reduces the measurement accuracy of the accelerometer. Therefore, removing the noise in the output signal of the accelerometer to improve its measurement accuracy has become a research hotspot.

Compared with improving the hardware structure of the accelerometer, denoising algorithms are widely used in noise removal of the accelerometer. Traditionally, Fourier transform, Kalman filtering (KF), Time-frequency peak filtering (TFPF) and other algorithms are often used for signal denoising [5,6,7,8]. These denoising algorithms have a good denoising effect but also have some inherent defects. For example, the time-domain positioning function of the Fourier transform is weak, and it cannot capture the change in instantaneous frequency with time very well; this makes it unsuitable for analyzing and processing non-stationary complex signals. Because of the matrix operation, the Kalman filter takes a long time to calculate and causes some waveform distortion [6]. TFPF is a signal enhancement technology that is widely used in seismic signal denoising and other fields [8,9,10,11]. For complex signals such as seismic signals, the signals are often nonlinear and non-stationary. In order to solve this problem, pseudo-Wigner–Ville distribution (PWVD) is adopted for denoising the noise signals locally. However, this also brings the problem of the window length selection of PWVD. As a tradeoff parameter, the length of the window has a certain influence on the signal denoising effect [9]. The long window has a strong denoising ability but tends to cause signal distortion. Although a short window can preserve useful information well, it causes insufficient signal removal.

Many hybrid denoising algorithms are proposed based on adaptive decomposition algorithms such as EMD and LMD and have been widely used. The experiment proves that the hybrid denoising algorithm can improve the original denoising algorithm to a certain extent and has a better effect [12,13,14,15]. Lu et al. [6] introduced the EMD and wavelet threshold hybrid denoising algorithm for the MEMS accelerometer, using EMD to decompose the output signals and obtain a series of IMFs, and then using wavelet threshold to denoise high-frequency IMFs; finally, the denoising signal is obtained by reconstruction, the hybrid denoising algorithm preserves the useful information of the signal to some extent while denoising. Li et al. [13] combined CEEMDAN with wavelet threshold denoising and applied it to the denoising of underwater acoustic signals and achieved good results. Ning et al. [11] introduced a joint denoising algorithm combining LMD and TFPF and applied it to the denoising of gearbox vibration signals. In this algorithm, after LMD decomposition, the experimental signals are decomposed into some product functions (PFs), then sample entropy is introduced to classify those PFs into the useful components, mixed components and noise components. The short-window TFPF and long-window TFPF are used for denoising the useful components and the mixed components, respectively, and then the noise component is discarded. This method improves window length selection for TFPF and has a good denoising effect. However, both EMD and LMD have inherent defects such as modal aliasing and a weak theoretical basis. In comparison, the VMD proposed by Dragomiretskiy et al. [16] has many advantages of solid theoretical basis, obvious decomposition effect and so on, and it is widely used in the engineering fields and has achieved good results [17,18,19,20,21,22,23].

Unfortunately, the proper decomposition parameters of VMD should be selected before use. When the decomposition number k is set unreasonably, the phenomenon of over-decomposition or under-decomposition occurs. On the other hand, the larger the penalty factor α, the wider the bandwidth of the intrinsic mode function, and vice versa, which affects the decomposition accuracy of VMD [23]. Therefore, it is significant to select the appropriate VMD decomposition parameter [*k*, *a*]. Thanks to the emergence of intelligent algorithms such as particle swarm optimization algorithms and neural network algorithms, many scholars have used these algorithms to optimize the VMD [24,25,26]. These optimization algorithms realize the purpose of optimization by constructing the single objective function, which only considers the problem in one aspect, while the multi-objective optimization algorithm comprehensively considers the optimization of the target from many elements and can obtain the optimal global characteristics. As one of the multi-objective optimization algorithms, the MOPSO [27] was successfully applied to the engineering fields in view of its simple theory, fast convergence, strong global optimization ability, flexible parameter adjustment mechanism and other characteristics [23].

In this paper, the improved VMD and TFPF were combined and applied in the denoising of the HGMA output signal. Firstly, the VMD was optimized by MOPSO, and the optimal decomposition parameters [*k_best_*, *a_best_*] could be searched, in which the permutation entropy (PE) and fuzzy entropy (FE) were selected for MOPSO as fitness functions. Secondly, after decomposition by the improved VMD, the HGMA output signal was decomposed into some IMFs. Then, these IMFs were classified into information-dominated or noise-dominated IMFs by sample entropy (SE). Information-dominated IMFs are mainly composed of useful signals mixed with a small amount of noise, while noise-dominated IMFs are mainly composed of noise with a small number of useful signals. Thirdly, we adopted short-window TFPF for information-dominated IMFs denoising, while long-window TFPF was adopted for noise-dominated IMFs denoising. Finally, the accelerometer denoising signal was obtained by reconstructing those denoised IMFs. The experimental results show that the improved VMD and TFPF hybrid denoising algorithm has a smaller signal distortion and stronger denoising ability, so it can be adopted to denoise the output signal of the High-G MEMS accelerometer to improve its accuracy. The structure of the rest is as follows: the second part introduces the basic principle of the improved VMD and TFPF, the third part introduces the HGMA, the fourth part is the simulation and experimental analysis, the fifth part is the analysis of the experimental results and the conclusion is given in the last part.

## 2. Algorithm Description

### 2.1. Variational Modal Decomposition (VMD)

The VMD is an effective decomposition method for processing non-stationary signals. Different from EMD and LMD, which decompose complex signals by recursion-filter decomposition, VMD decomposes complex signals by non-recursive and variational mode decomposition. The optimal solution of the variational model is searched through cyclic iterative processing, which means that the complex signals are decomposed into many intrinsic mode functions (IMFs), and each IMF has the center frequency and limited bandwidth. This enables VMD to avoid the mode aliasing phenomenon existing in EMD and LMD and has better noise robustness. The decomposition principle of VMD is briefly described as follows [16].

1. The construction of the constrained variational model.

Suppose that any complex signal *y*(*t*) is decomposed into k IMFs {*u_k_*(*t*)} = {*u*_1_(*t*), *u*_2_(*t*), *u*_3_(*t*), …, *u_k_*(*t*)} with center frequency and finite bandwidth. The variational model is constructed to seek the optimal modal functions so as to minimize the sum of estimated bandwidths of all intrinsic mode functions. The variational model is constructed as follows:

a. Hilbert transformation is performed on the obtained mode functions to obtain their analytic signals; the purpose is to obtain the unilateral spectrum of each mode function:(1)$$\left[\sigma (t)+\frac{j}{\pi t}\right]*u_{k}(t)$$

b. To obtain the constrained variational model, the center frequency of each modal analytical signal obtained in Formula (1) is initialized, then the square norm of demodulation signal gradient is calculated, and the bandwidth of each IMF is estimated:(2)$$\{\begin{array}{c} \underset{\{u_{k},\theta_{k}\}}{\min}\{\sum_{k}{\left\|\partial_{t}\left[\left(\sigma (t)+\frac{j}{\pi t})u_{k}(t\right)\right]e^{-j\theta_{k}t}\right\|}_{2}^{2}\}\\ s.t\sum_{k}u_{k}=y(t) \end{array}$$
where, {*θ_k_*} = {*θ*_1_, *θ*_2_, …, *θ_k_*} is the collection of central frequencies of each IMF.

2. The solution of the constrained variational model.

a. To simplify the constrained variational model, the unconstrained variational model is constructed by constructing an extended Lagrangian expression. In Equation (3), *α* and *λ* are the penalty factor and Lagrangian multiplication operator.
(3)$$\begin{array}{l} L\left(\{u_{k}\},\{\theta_{k}\},\lambda \right)=\alpha {\sum_{k}\|\partial_{t}\left[\left(\sigma (t)+\frac{j}{\pi t}\right)*u_{k}(t)\right]e^{-j\theta_{k}t}\|}_{2}^{2}\\ +{\|y(t)-\sum_{k}u_{k}(t)\|}_{2}^{2}+\langle \lambda (t),y(t)-\sum_{k}u_{k}(t)\rangle \end{array}$$

b. The corresponding extremum solution can be obtained by transforming the Lagrangian function obtained by Formula (3) in the time-frequency domain. The expressions for the *u_k_* and *θ_k_* are as follows, respectively:(4)$$u_{k}^{n+1}(\theta )=\frac{y-\sum_{i\neq k}^{}u_{i}(\theta )+\frac{\lambda (\theta )}{2}}{1+2\alpha {\left(\theta -\theta_{k}\right)}^{2}}$$
(5)$$\theta_{k}^{n+1}=\frac{\int_{0}^{\infty }\theta |u_{k}(\theta ){|}^{2}d\theta }{\int_{0}^{\infty }\left|u_{k}(\theta \right){|}^{2}d\theta }$$

c. The alternating direction multiplier algorithm is adopted to update the parameters *u_k_^n^*^+1^, *θ_k_^n^*^+1^ and *λ^n^*^+1^, and the update formula of *λ^n^*^+1^ is:(6)$$\lambda^{n+1}(\theta )\leftarrow \lambda^{n}(\theta )+\tau [y(\theta )-\sum_{k}u_{k}^{n+1}(\theta )]$$

In Equation (6), *τ* is the time constant factor, which affects the update of *λ*. If the accuracy is not strictly required, the update can be avoided. In this case, *τ* = 0.

d. When the condition of Formula (7) is satisfied, the iteration stops and *k* intrinsic mode functions are output. Otherwise, the iteration continues by following the formulas above.
(7)$$\sum_{k=1}^{K}\frac{{\|u_{k}^{n+1}(w)-u_{k}^{n}(w)\|}_{2}^{2}}{{\|u_{k}^{n}(w)\|}_{2}^{2}}<\varepsilon$$

Although VMD has a good decomposition effect, it also has an inherent defect; thus, it needs to rely on experience to set the decomposition parameters [*k*, *a*] before the decomposition. Improper setting of decomposition parameters affects the decomposition performance of VMD, so it is necessary to optimize the VMD.

### 2.2. Parameter Optimization of VMD Based on MOPSO

The MOPSO algorithm is a widely used intelligent algorithm that combines particle swarm optimization (PSO) and the grid algorithm; it advances the original single target optimization to multiple targets. It is based on the predation behavior of birds, and it has excellent convergence speed and good overall search ability. Reasonable selection of multiple fitness functions is also the key to MOPSO; the fuzzy entropy (FE) and permutation entropy (PE) were selected as fitness functions of MOPSO to optimize the VMD in this article.

A brief introduction to Fuzzy entropy (FE) is as follows [28]:

Fuzzy entropy (FE) is an algorithm that can judge the complexity of the measured complex nonlinear signal by calculating the probability of the new mode generated in the time series. At the same time, FE is also an improvement on the approximate entropy and sample entropy, which overcomes the situation that their entropy value is not continuous in the extraction process. In addition to inheriting the advantages of the first two kinds of entropy, fuzzy entropy is less dependent on time series and more robust to noise-containing signals. Therefore, FE was selected for MOPSO as one of the fitness functions in this article.

The steps of fuzzy entropy (FE) are as follows:

Step.1 Reconstruct phase space

For the time series {*s*(*p*), 1 ≤ *p* ≤ *N*}, phase space reconstruction is carried out to obtain m-dimensional vectors:(8)$$\begin{array}{l} X_{p}^{m}=\{s(p),s(p+1),\ldots ,s(p+m-1)\}-s_{0}(p)\\ \begin{array}{cc} & p=1,2,\ldots ,N-m+1 \end{array} \end{array}$$

Here, *X_p_^m^* is m consecutive values of s starting at the *pth* point and subtracting the mean *s*_0_(*p*),
(9)$$s_{0}(p)=\frac{1}{m}\sum_{q=0}^{m-1}s(p+q)$$

Step.2 Define the distance between vectors.

*D^m^_pq_* is the maximum difference between vector *X^m^_p_* and *X^m^_q_*, namely:(10)$$\begin{array}{l} D_{pq}^{m}=d\left[X_{p}^{m},X_{q}^{m}\right]=\underset{k\in (0,m-1)}{\max}\{|[s(p+k)-s_{0}(p)]-[s(k+q)-s_{0}(q)]|\}\\ (p,q=1,2,\ldots ,N-m,p\neq q) \end{array}$$

Step.3 Compute the membership degree between vectors.

The membership degree of vector *X^m^_p_* and *X^m^_q_* is defined as *µ*(*d^m^_pq_*, *θ*, *ω*), which is:(11)$$D_{pq}^{m}=\mu (d_{pq}^{m},\theta ,\omega )=e^{-{\left(\frac{d_{pq}^{m}}{\omega }\right)}^{\theta }}$$

In the formula, the fuzzy function is defined as *µ*(*d^m^_pq_*, *θ*, *ω*), which is an exponential function, and the gradient and width of its boundary are denoted as *θ* and *ω*.

Step.4 Define the function.
(12)$$\Phi^{m}(\theta ,\omega )=\frac{1}{N-m}\sum_{p=1}^{N-m}(\frac{1}{N-M-1}\sum_{q=1,q\neq p}^{N-m}D_{pq}^{m})$$

Similarly, for *m* + 1 dimension vector, repeat the Formulas (8)–(11); the formula can be obtained:(13)$$\Phi^{m+1}(\theta ,\omega )=\frac{1}{N-m}\sum_{p=1}^{N-m}(\frac{1}{N-M-1}\sum_{q=1,q\neq p}^{N-m}D_{pq}^{m+1})$$

Step.5 Define fuzzy entropy.
(14)$$F_{E}(m,\theta ,\omega )=\underset{N\to \infty }{\lim}[\ln\Phi^{m}(\theta ,\omega )-\ln\Phi^{m+1}(\theta ,\omega )]$$

When *N* is a finite value, the Equation (14) is simplified as follows:(15)$$F_{E}(m,\theta ,\omega )=\ln\Phi^{m}(\theta ,\omega )-\ln\Phi^{m+1}(\theta ,\omega )$$

Another fitness function, Permutation entropy (PE), is introduced as follows:

PE was first proposed by Bandt et al. [29], which can be used to calculate the complexity and randomness of complex signals; the principle of PE is as follows:

Step.1 Reconstruct phase space

For the time series {*u*(*j*), 1 ≤ *j* ≤ *N*}, phase space reconstruction is carried out to obtain a phase sequence:(16)$$R=\left[\begin{array}{c} R(1)\\ R(2)\\ \vdots \\ R(l)\\ \vdots \\ R(k) \end{array}\right]=\left[\begin{array}{cccc} u(1) & u(1+\omega ) & \ldots & u(1+(e-1)\omega )\\ u(2) & u(2+\omega ) & \ldots & u(2+(e-1)\omega )\\ \vdots & \vdots & & \vdots \\ u(l) & u(l+\omega ) & \ldots & u(l+(e-1)\omega )\\ \vdots & \vdots & & \vdots \\ u(k) & u(k+\omega ) & \ldots & u(k+(e-1)\omega ) \end{array}\right]$$

Here *e* is the embedded dimension, *k* + (*e* − 1)*ω* = *n*, *R*(*l*) represents the reconstructed vector, there are a total of *k* reconstruction vectors, and the delay time is denoted as *ω*.

Step.2 Rearrange the reconstructed vectors.

Each reconstructed vector is rearranged according to the size, then the column indexes of elements in the vector are obtained to form a set of symbol sequences {*h*_1_, *h*_2_, *h*_3_…, *h_m_*}, namely:(17)$$s(l+(h_{1}-1)\omega )\leq \ldots \leq s(l+(h_{m}-1)\omega )$$

When *h_p_* < *h_q_*, that is:(18)$$s(l-(h_{p}-1)\omega )\leq s(l-(h_{q}-1)\omega )$$

Step.3 The calculation and normalization.

After the rearrangement, calculate the probability of each symbol sequence and denote them as P_1_, P_2_…, P_r_, and the calculation formula of permutation entropy is:(19)$$H_{p}(e)=-\sum_{n=1}^{e}p_{k}\ln p_{k}$$

The maximum of permutation entropy is ln *e*!, normalize the permutation entropy, which is:(20)$$H_{p}=\frac{H_{p}(e)}{\ln e!}\begin{array}{ccc} & & H_{p}\in [0,1] \end{array}$$

The normalized permutation entropy can be used to calculate the complexity and randomness of complex signals; thus, the larger the permutation entropy is, the higher the complexity and randomness of complex signals are, and vice versa.

The fuzzy entropy (FE) and permutation entropy (PE) were selected for MOPSO as the fitness functions to optimize the VMD. A brief description of the steps of the MOPSO algorithm is as follows [23]:

A. Firstly, key parameters of MOPSO are set, including total particle number N_P_, maximum iteration number M, save set size N_R_, etc. The number of particles affects the searching ability of MOPSO. When the number of particles is set too large, the algorithm has a good global searching ability, but it affects the speed of the algorithm.

B. Initialize the particle swarm P1: The position P(*j*) of each particle is randomly initialized, while its velocity v(*j*) is set to zero. The fuzzy entropy and permutation entropy are adopted as fitness functions to evaluate each particle. When the fitness values are smaller, the corresponding parameters are better. The non-inferior solution in P_1_ is stored in the save set N_P_.

C. Update the individual best particle P_best_ and the global best particle G_best_, use the adaptive grid method to find the global optimal particle G_best_, and continuously evolve to generate the next generation particle population; perform the following steps before the save set reaches the maximum:

a. Calculate the density information of the particles in the save set, divide the target space into small areas by the grid, and the density is measured by the number of particles in each area;

b. The optimal historical position is updated when the particle’s current position is better than the best position of the individual history. Then, the global optimal particle G_best_ is selected for the particles in the population, and the selection is based on the density information of the particles. Specifically, for a particle in a save set, the lower the density value, the greater the probability of selection;

c. Update the position and velocity of each particle. In addition, the particles search for the optimal solution under the leadership of G_best_ and P_best_:(21)$$v_{i,d+1}^{j}=\mu (wv_{i,d}^{j}+c_{1}R_{1}(P_{i,d}^{j}-x_{i,d}^{j})+c_{2}R_{2}(G_{i,d}^{j}-x_{i,d}^{j}))$$
(22)$$x_{i,d+1}^{j}=x_{i,d}^{j}+v_{i,d+1}^{j}$$
where *d* represents the algebra of the current particle evolution, *i* represents the current evolutionary particle, *c*_1_ and *c*_2_ are the learning factors, *μ* is the contraction factor, *R*_1_ and *R*_2_ are the random numbers in the interval [0, 1], $P_{i,t}^{j}$ and $G_{i,t}^{j}$ represent the value of the *j*-th decision vector of P_best_ and G_best_ of the particle, respectively. The save set is updated after the evolution of the new generation group P_t+1_; the non-inferior solutions in P_t+1_ are saved to the save set.

D. If the number of particles in the save set exceeds the set maximum value, the individuals in the dense range are replaced, and the individuals in the sparse range are retained to maintain the size of the save set. For a grid with more than one particle, calculate the number of particles *ND* to be deleted in the grid according to Formula (23), and then randomly delete *ND* particles in the grid.
(23)$$ND=Int(\frac{\left|A_{t+1}-\bar{N}\right|}{\left|A_{t+1}\right|}\times Grid[k,2]+0.5)$$
where *A_t_* is the number of particles in the save set, *Gird*[*k*] is the number of particles in grid *k*.

E. When the stop condition is reached, the iteration is stopped, the particle information in the storage set is output, and the optimal decomposition parameters [*k_best_*, *a_best_*] can be obtained. The flow diagram of the improved VMD is given in Figure 1.

### 2.3. Time-Frequency Peak Filtering (TFPF)

Time-frequency peak filtering is a noise elimination technology proposed by Mesbah et al. [30]. It has been applied in many engineering fields widely due to its ability to extract effective signals in a noisy environment.

TFPF algorithm is mainly based on the Wigner–Ville distribution (WVD) and instantaneous frequency estimation theory to filter and denoise signals. Due to its good time-frequency focusing property, WVD is widely used in engineering. However, when WVD processes multi-component signals, the resolution of the time-frequency distribution of signals is reduced due to the generation of cross terms, which leads to the weakening of the time-frequency focusing of WVD. In order to improve TFPF, the pseudo-Wiener–Ville distribution (PWVD) is used to suppress the cross terms. According to the principle of TFPF, it is necessary to encode the noisy signal to make it become the analytic signal of instantaneous frequency firstly, and the estimated value of the effective signal can be obtained through the instantaneous frequency estimation.

The output signal of the accelerometer is mixed with noise, which is:(24)y(t)=o(t)+n(t)

Here, *o*(*t*) and *n*(*t*) represent the vibration signal and random noise in the accelerometer output signal, respectively.

A brief introduction to TFPF is as follows [10]:

Step 1. The frequency modulation is carried out for the signal *y*(*t*), and the analytic signal *h*(*t*) is obtained:(25)$$h(t)=e^{j2\pi \mu \int_{0}^{t}y(\lambda )d\lambda }$$

Here, *µ* is the frequency modulation index.

Step 2. The PWVD spectrum of the analytic signal *h*(*t*) is calculated:(26)$$PW_{2}(t,f)=\int_{-\infty }^{\infty }w(\tau )h(t+\frac{\tau }{2})h^{*}(t-\frac{\tau }{2})e^{-j2\pi ft}d\tau$$
where *t* stands for time, *τ* stands for integral variable, *f* stands for frequency, *h** stands for the conjugated operator of *h*, the window function is denoted as *w*(*τ*), and the window length is a tradeoff parameter of TFPF.

Step 3. According to the maximum likelihood estimation principle, the peak value of the PWVD spectrum of the analytic signal is calculated to estimate the instantaneous frequency of the analytic signal, then the amplitude estimation of the original effective signal is obtained:(27)$$f_{z}(t)=\frac{\arg\max[PW_{z}(t,f)]}{\mu }$$

The window length in the TFPF algorithm is the key parameter that affects the denoising effect. The window length directly determines the signal fidelity and noise removal effect. When selecting a long window for denoising, the noise of the signal can be eliminated more cleanly, but the amplitude of the signal is reduced, resulting in attenuation of useful signals, especially at the peak and trough of the wave. On the contrary, selecting a short window for denoising ensures the retention of useful signals, but it is deficient in noise suppression, and there are still more noise components left after filtering. Therefore, it is inappropriate to simply adopt a long window or short window to denoise the whole signal.

### 2.4. Introduction of the Improved VMD and TFPF

In order to improve TFPF in the selection of the window length, this article combines VMD with TFPF. In order to achieve a better decomposition effect, MOPSO is adopted to optimize VMD. In addition, the idea of classification processing is adopted, and the sample entropy (SE) is introduced to classify IMFs. Figure 2 is the flow chart of the improved VMD and TFPF hybrid denoising algorithm, and the steps are as follows:

Step 1. Optimization of VMD decomposition parameters.

Before VMD decomposition, the decomposition parameters *k* and *a* must be determined empirically, which may easily lead to the fact that the decomposition results do not conform to the actual situation. Therefore, the MOPSO is adopted to optimize VMD in this article. The flow diagram of the improved VMD is given in Figure 1.

Step 2. Variational modal decomposition.

After determining the decomposition parameters [*k_best_*, *a_best_*], the output signal of the accelerometer is decomposed by VMD, and a series of IMFs are obtained. For each IMF, it is neither pure signal nor pure noise, but generally, the mixed component of noise and useful signals.

Therefore, we introduce SE as a judgment criterion to classify these IMFs.

Step 3. Calculation and classification.

In order to make denoising more targeted, this paper introduces SE to distinguish IMFs. By calculating the sample entropy, the IMFs can be classified into noise-dominated or information-dominated IMFs. The noise-dominated IMFs are composed of a large amount of noise and a small number of useful signals; the noise should be eliminated to make it as clean as possible. For the information-dominated IMFs components, they are mainly composed of useful signals with a small amount of noise, and those useful signals need to be well preserved.

Step 4. Targeted denoising and reconstruction.

According to the different characteristics of each IMF, TFPF with different window lengths should be selected for denoising. Since the long-window TFPF has good noise elimination characteristics, it is adopted to deal with the noise-dominated IMFs. The short-window TFPF has a relatively weak denoising effect but little signal distortion, so it can be adopted to deal with information-dominated IMFs, which contain a little noise. Finally, we obtain the accelerometer denoising signal by restructuring those denoised IMFs.

## 3. High-G MEMS Accelerometer (HGMA)

The experimental signals collected in this article are from a High-G MEMS accelerometer (HGMA). The HGMA works according to the piezoresistive effect and has good effects on the aspects of survivability under high impact and high range. In terms of structure, this accelerometer adopts the four-beam and central-island mass style. To make the manufacturing simple, the beam and mass of the accelerometer are rectangular, fixed by the frame and connected to the base of the accelerometer; the structure and mechanical model of HGMA is shown in Figure 3 and Figure 4, respectively.

As shown in the mechanical model of HGMA, the coordinate system could be established according to the accelerometer cross-section, and then we could follow the right-hand rule to determine the coordinate axis. The center axis of symmetry is the *Z*-axis, and its positive direction is down, and for the *X*-axis, its positive direction is to the right. In addition, the structural parameters of beam and mass and their specific values are given in Table 1.

When the accelerometer operates in different modes, the sensitivity and bandwidth of the accelerometer are affected. Therefore, ANSYS was adopted to analyze the first four modes of the accelerometer. The simulation results are given in Figure 5.

The structural deformation can be observed from the simulation results. Among them, the first mode is the working mode of the accelerometer (it can be observed from Figure 5a that the HGMA’s mass vibrates along the *Z*-axis while its frame remains stationary, and the surface of the mass is parallel to the *X*-*Y* plane). In this mode, the resonant frequency of the accelerometer is 408 khz, and it can provide a wide test bandwidth. In Figure 5b,c, the second and third modes of the accelerometer are flipped along the *X*-axis and *Y*-axis. In addition, the resonant frequencies of the two modes are 667 khz and 671 khz, respectively. The fourth mode of the accelerometer is given in Figure 5d; in this mode, the HGMA’s frame and mass vibrate along the *Z*-axis. In addition, the resonant frequency of the fourth modes is 119 khz.

The HGMA is fabricated based on silicon and glass, and the prototype has a piezoresistor error of less than 1%, a sensitivity error of less than 15%, a range over 100,000 g, and a sensitivity of 0.5611 μV/g. From −10 °C to 60 °C, the bias of HGMA varies by 8.5%. The fabrication and test process are shown in [3].

Additionally, HGMA’s SEM and CCD images are shown in Figure 6.

## 4. Simulation and Experimental Analysis

In order to verify the feasibility of the improved VMD, the vibration signal containing multiple modulation sources is constructed for testing.
(28)$$\{\begin{array}{l} y(t)=s_{1}(t)+s_{2}(t)+s_{3}(t)+2\times randn(t)\\ s_{1}(t)=\sin(2\pi f_{1}t)\\ s_{2}(t)=(1+\cos(2\pi f_{2}t)+\cos(2\pi f_{3}t))\times \sin(2\pi f_{4}t)\\ s_{3}(t)=A_{m}\times \exp(-g/-T_{m})\times \sin(2\pi f_{5}t) \end{array}$$
where *s*_1_(*t*) is a sinusoidal signal, *s*_2_(*t*) is an analog signal with two modulation sources, whose modulation frequencies are *f*_2_ and *f*_3_, respectively, and *f*_4_ is the carrier frequency. In addition, *s*_3_(*t*) is the periodic impact signal; *A_m_*, *g*, *T_m_* and *f*_5_ represent the impact amplitude, the damping coefficient, the impact period and the rotation frequency, respectively; and these main parameters of the constructed vibration signal are given in Table 2. And the time-domain waveforms of the constructed signals are shown in Figure 7.

The key parameters of MOPSO are set as [23]:

The population size Np and the save set size NR are set at 30; the maximum iteration number M is set at 10, the inertia weight W is set at 0.4, while the learning factors c1 and c2 are both set at 1.5. Firstly, the signal is decomposed by VMD, then calculate the permutation entropy and fuzzy entropy of each IMF are obtained by VMD decomposition, and then find the minimum values of permutation entropy and fuzzy entropy, respectively, and taken as fitness functions 1 and 2.

Figure 8 shows the solution set distribution of VMD parameter optimization based on MOPSO, the particles marked in red are the Pareto frontier optimal solutions. Finally, the optimal decomposition parameter [*k_best_*, *a_best_*] = [3, 2187] is found, and the VMD algorithm parameters k = 3, α = 2187 are set, then the constructed signal is de-composed by the improved VMD. To intuitively evaluate the decomposition effect of the improved VMD, this article is compared with the EMD algorithm.

Figure 9 shows the decomposition of EMD. It can be seen that 10 IMFs are obtained after EMD decomposition. From its frequency domain, only the first few IMFs are meaningful. The frequency component of 235 Hz is decomposed into IMF1 and IMF2, resulting in mode aliasing. IMF3 is meaningless, and the frequency component of 55 Hz can be extracted from IMF4, while the frequency component of 280 Hz cannot be extracted. Figure 10 shows the decomposition of VMD. The improved VMD in this paper decomposes the signal into three IMFs. In IMF1, the 55 Hz low-frequency component in the original signal is extracted; in IMF2, the 235 Hz central frequency and the 10 Hz sideband evenly distributed on the two sides are also obvious; and in IMF3, the 500 Hz high-frequency component in the original signal is extracted. Through comparison, it can be found that the frequency extraction is not accurate due to the problems of EMD mode aliasing, while the improved VMD method can extract useful information from strong background noise and has a good decomposition ability.

The decomposition effect of the improved VMD is verified in the previous simulation experiment. Next, we applied the improved VMD combined with TFPF to the denoising of HGMA output signals.

The HGMA is calibrated by the Hopkinson Bar calibration system in the experiment, and the HGMA’s output voltage signals are collected for further analysis. The entire calibration device is shown in Figure 11, which consists of the Recycling Box, Deformeter, Computer, Hopkinson Bar and Compressed Air. The working voltage required by HGMA is provided by the power supply (GwinstekGPS-4303C), and the high-speed acquisition system is used to collect the accelerometer voltage signal. The ambient temperature of the whole experiment is 25 °C, and the sampling points are 19243.

Figure 12 shows the output signal of the HGMA after calibration. The whole experimental signal is divided into three parts: the static phase, the shock phase and the vibration phase. Due to the influence of the experimental environment, amplifier and HGMA itself and other factors, there are relatively obvious noises before the impact (static phase), and these noises accompany the whole calibration test process. In order to improve the calibration accuracy, the noises need to be removed. The impact phase is the period of time during which the Hopkinson bar impact produces the first acceleration signal. The vibration phase is mainly due to the vibration output of the HGMA sensor driven by the calibration device, and the vibration frequency in this phase is approximately about 500 kHz. Among them, the shock and vibration phases are important parts of the calibration experiment because these two parts can reflect the dynamic characteristics of the output signal of the accelerometer. However, the existence of noise causes measurement error, so it is necessary to remove the “burr” attached to the calibration signal.

According to the algorithm steps, the experimental signals need to be decomposed by VMD first. Similarly, the optimal parameters [*k_best_*, *a_best_*] need to be determined before decomposition. The parameters for MOPSO are set as follows: Np and NR are set at 30, the maximum iteration number M is set at 10, the inertia weight W is set at 0.4, while the learning factors c_1_ and c_2_ are both set at 1.5. The optimization range of parameters k and a are set as [4, 12] and [1000, 5000], respectively. The optimization situation is shown in Figure 13. Figure 13a is the particle distribution of the last iteration, where the Pareto front optimal solution set is marked in red, and Figure 13b is the optimization result, the optimal decomposition parameter [*k_best_*, *a_best_*] = [9, 4895].

The decomposition of VMD is given in Figure 14, the output signal is decomposed into nine IMFs, and then SE is adopted to distinguish these IMFs. In Figure 15, we can clearly see that these IMFs are classified into two categories. IMFs whose sample entropy value ranges mainly from 0 to 0.1 are considered information-dominated IMFs (IMF1, IMF2, IMF3, IMF4, IMF5, IMF9), and short-window TFPF is adopted for denoising. IMFs with sample entropy of 0.4~0.6 (IMF6, IMF7, IMF8) are considered noise-dominated IMFs, and long-window TFPF is adopted for denoising.

We implemented short-window TFPF and long-window TFPF denoising for IMFs dominated by useful signals and noise, respectively, and the denoising results are given in Figure 16. For the information-dominated IMFs, the signals before and after denoising remain basically the same, which largely preserves the useful information of the original signals. As for the noise-dominated IMFs, it can be seen that the long-window TFPF can remove the noise component well and obtain a relatively clean signal. The final HGMA denoising signal can be obtained by reconstructing those IMFs denoised by short-window or long-window TFPF.

## 5. Discussion

In order to highlight the performance of the improved VMD and TFPF denoising algorithm, we compared the improved VMD and TFPF with the EMD denoising algorithm and TFPF denoising algorithm. The comparison results are shown in Figure 17. The following discusses the denoising ability and signal loss of the algorithm from time and frequency domains.

**For the static phase:** The signal in the static stage contains abundant noise, and the peak-to-peak value is about 0.054 V. From the denoising results, it can be seen that these denoising methods all have good denoising effect. However, compared with EMD, the improved VMD-TFPF and TFPF denoising algorithms have stronger denoising abilities. After denoising with the improved VMD-TFPF algorithm, the peak-to-peak value of the signal is reduced to about 0.006 V.

**For the shock phase:** This stage is the main section of the accelerometer’s calibration, and its peak value is about −1.754V. In this stage, the denoising signals of the three denoising algorithms almost overlap with the original signal, which indicates that the three denoising algorithms can retain the useful information of the signals well when denoising.

**For the vibration phase:** The vibration phase mainly reflects the accelerometer’s dynamic characteristics. By comparing signal distortion caused by different algorithms, it is easy to find that the signal distortion after EMD denoising is more serious; while the amplitude loss of TFPF denoising signal is about 0.2 V, the amplitude loss of improved VMD-TFPF denoising signal is only about 0.05 V. In comparison, the improved VMD-TFPF denoising method has the minimum signal distortion and is more suitable for accelerometer denoising.

The spectrum diagram before and after denoising is shown in Figure 18, in which the “vibration stage” is amplified, and the peak frequency of the vibration stage is about 536 kHz. The results show that the amplitude and shape of the signal after denoising by the improved VMD-TFPF are more consistent with the original data. While the amplitude of the signal denoised by TFPF is distorted to a certain extent, the amplitude and waveform of the EMD denoising signal are both distorted seriously. The amplitude of the original signal is 0.249 V, the amplitude of the improved VMD-TFPF denoising is about 0.239 V, and the amplitude of the signal denoised by TFPF is about 0.201 V. The comparison results reflect that the improved VMD-TFPF denoising algorithm can better reflect the dynamic characteristics of HGMA.

In order to quantitatively analyze the denoising performance of different denoising methods, we also chose the signal-to-noise ratio (SNR) and root-mean-square error (RMSE) as indicators to evaluate the performance of the denoising algorithms. The calculated results are given in Table 3. The improved VMD and TFPF denoising algorithms have the highest SNR and the lowest RMSE, which are better than the other two denoising algorithms and more suitable for accelerometer denoising.

## 6. Conclusions

In this article, the improved VMD and TFPF are proposed to denoise the HGMA’s output signal. The MOPSO is used to optimize the VMD, then the optimal decomposition parameters [*k*, *a*] are determined. The intrinsic mode functions (IMFs) obtained from VMD decomposition can be classified into information-dominated IMFs or noise-dominated IMFs by Sample entropy (SE). The information-dominated IMFs are denoised by short-window TFPF, and the noise-dominated IMFs are denoised by long-window TFPF. The denoising results of different denoising algorithms in the time domain and frequency domains were compared, and SNR and RMSE were taken as denoising indicators. The experimental results show that the improved VMD and TFPF denoising method has the smaller signal distortion, stronger denoising ability, the highest SNR and lowest RMSE, so it can be adopted to denoise the output signal of the High-G mems accelerometer to improve its accuracy.

## Figures and Tables

**Figure 1 micromachines-13-00891-f001:**
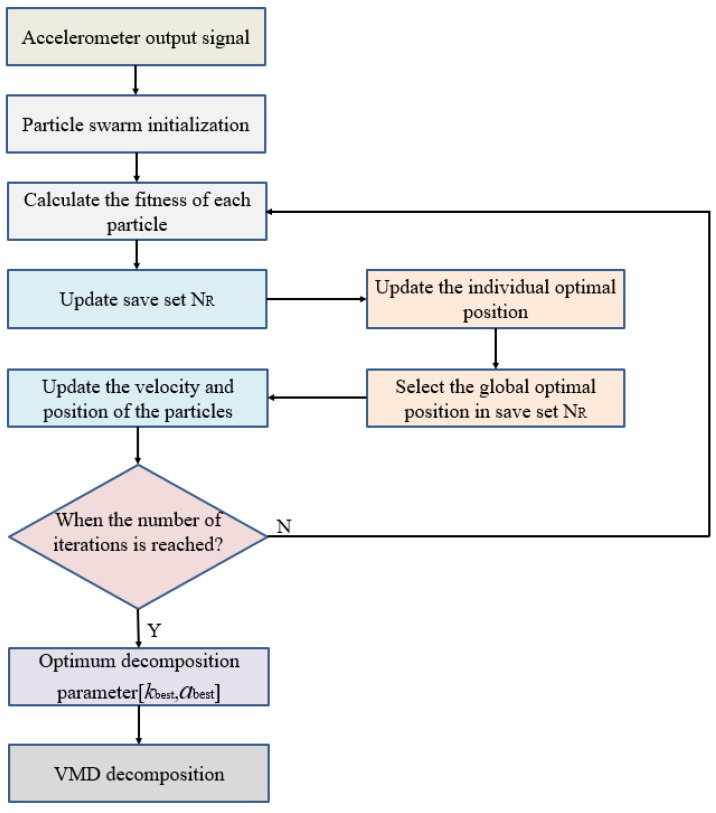
The flow diagram of the improved VMD.

**Figure 2 micromachines-13-00891-f002:**
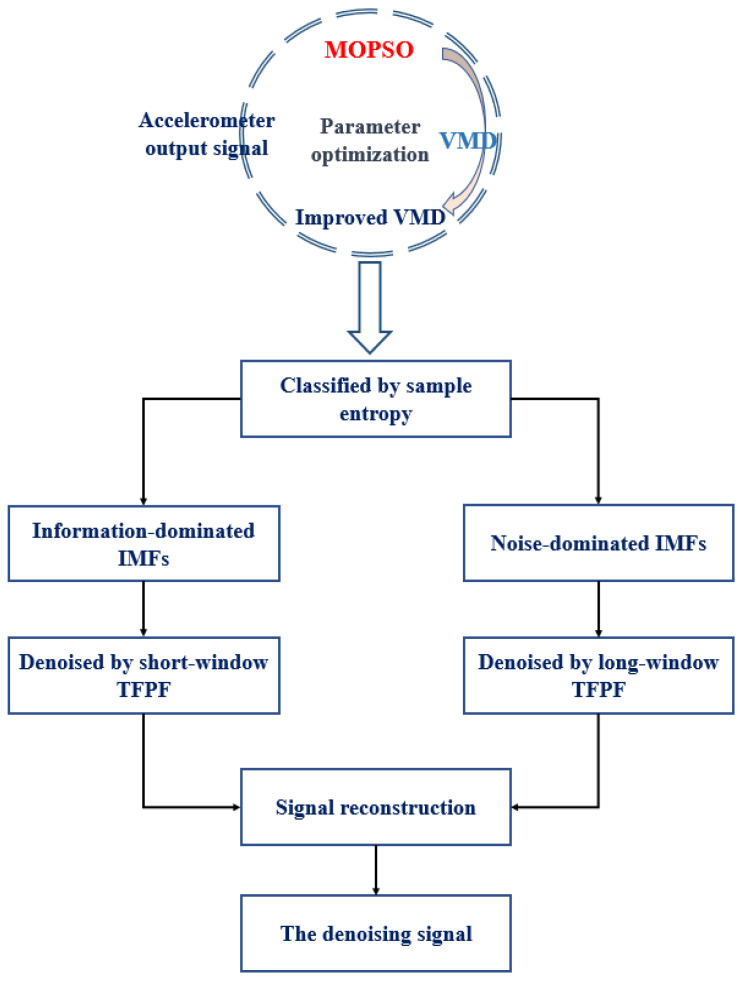
The flow diagram of the improved VMD and TFPF hybrid denoising algorithm.

**Figure 3 micromachines-13-00891-f003:**
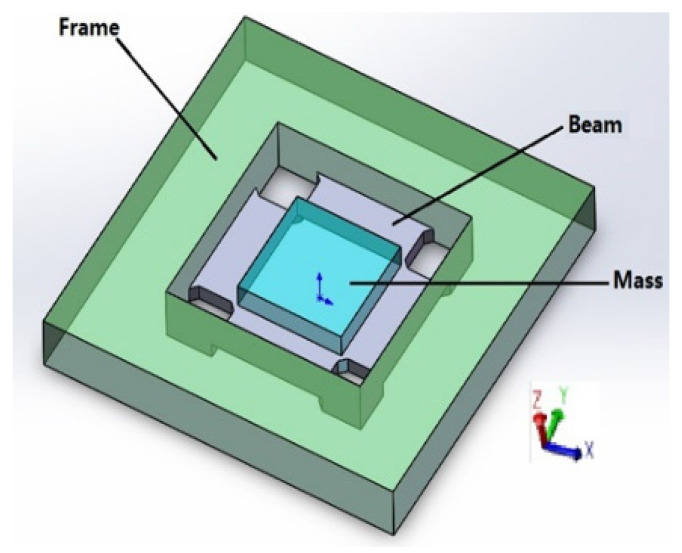
HGMA’s structure.

**Figure 4 micromachines-13-00891-f004:**
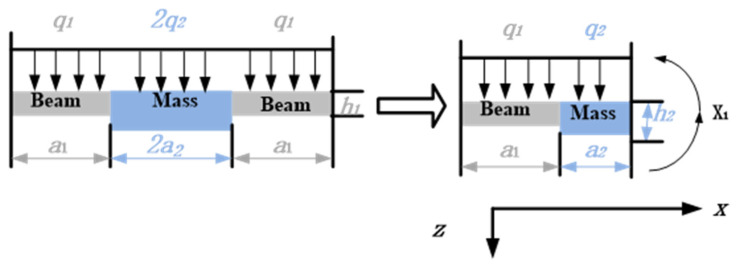
HGMA’s mechanical model.

**Figure 5 micromachines-13-00891-f005:**
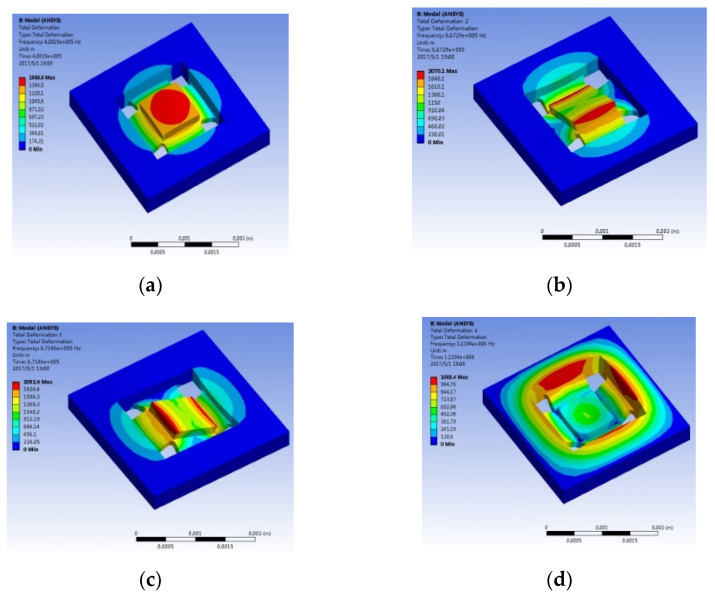
The lowest four modes of HGMA, (**a**) first mode shape, working mode; (**b**) second mode shape, disturbing mode; (**c**) third mode shape, disturbing mode; (**d**) fourth mode shape, disturbing mode.

**Figure 6 micromachines-13-00891-f006:**
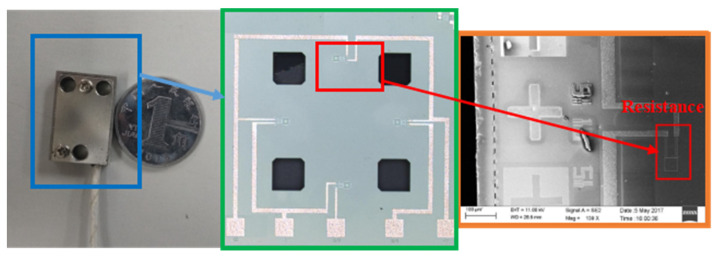
HGMA’s Overall photo, CCD and SEM photos.

**Figure 7 micromachines-13-00891-f007:**
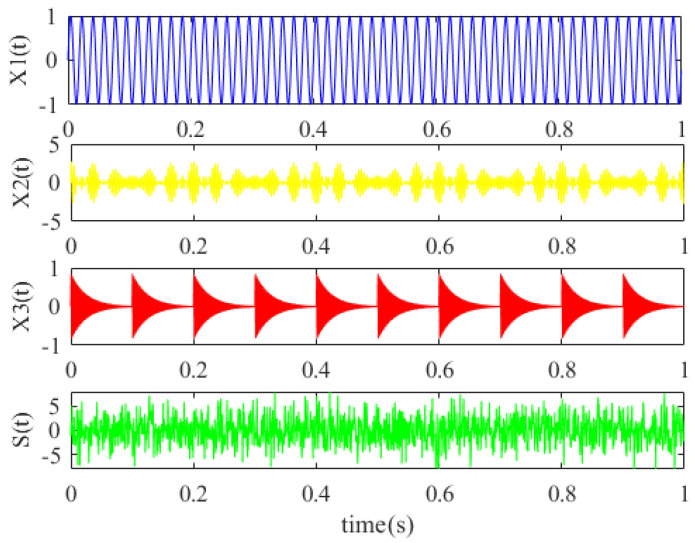
The time-domain waveforms of the constructed signals.

**Figure 8 micromachines-13-00891-f008:**
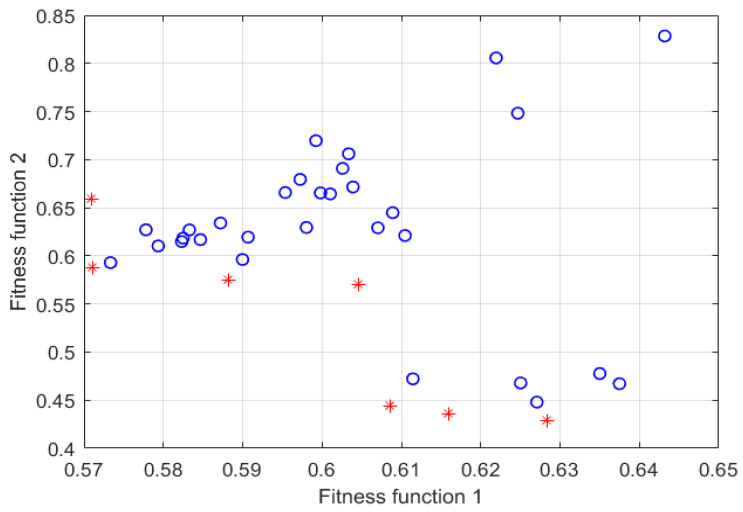
The solution set of VMD parameter optimization based on MOPSO. (where, o represents the non-dominant solution set, and * represents the Pareto optimal solution set).

**Figure 9 micromachines-13-00891-f009:**
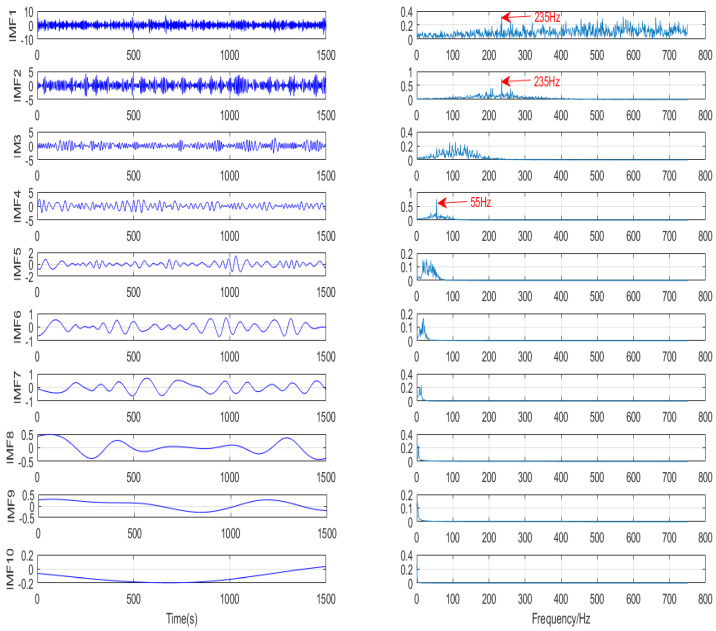
EMD decomposition results.

**Figure 10 micromachines-13-00891-f010:**
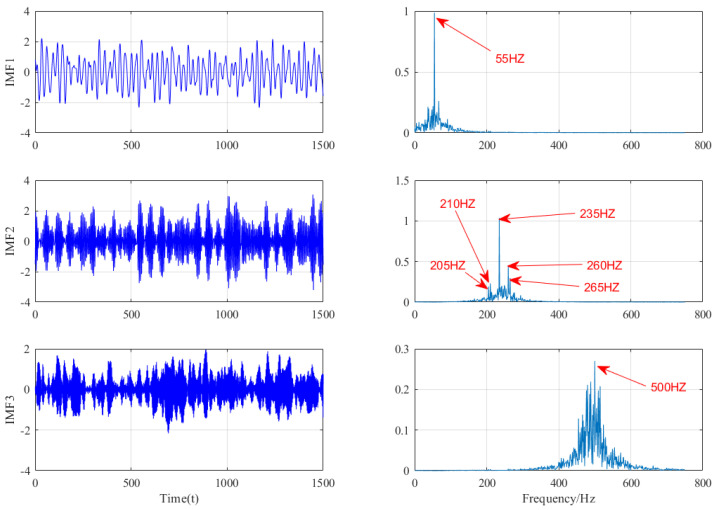
VMD decomposition results.

**Figure 11 micromachines-13-00891-f011:**
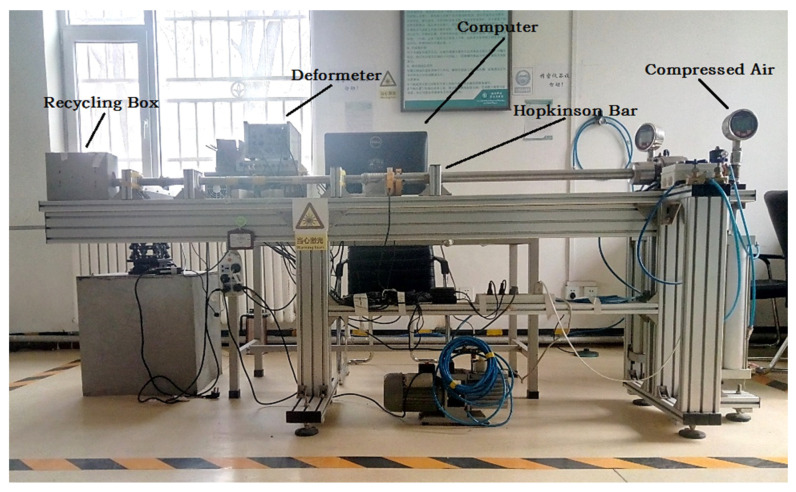
Hopkinson Bar calibration system.

**Figure 12 micromachines-13-00891-f012:**
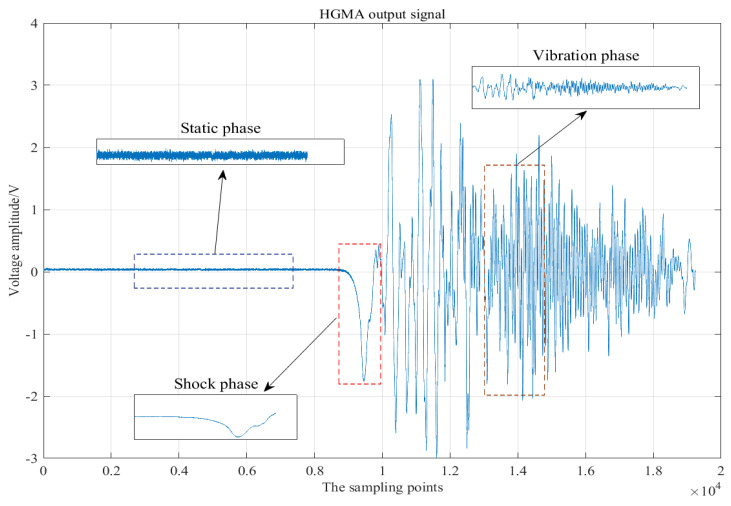
Voltage output signal of HGMA.

**Figure 13 micromachines-13-00891-f013:**
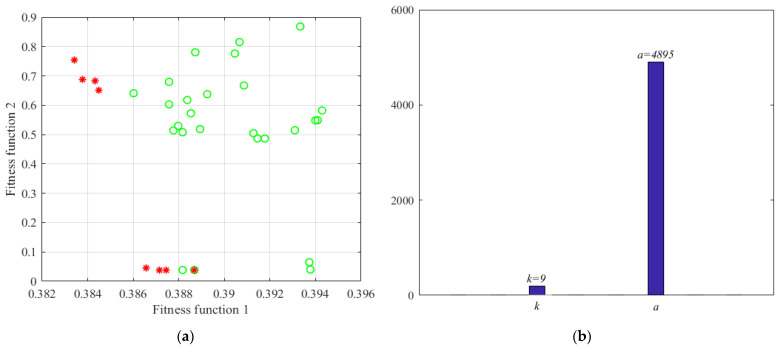
The optimization results of VMD, (**a**) is the particle distribution of the last iteration; (**b**) is the optimization result.

**Figure 14 micromachines-13-00891-f014:**
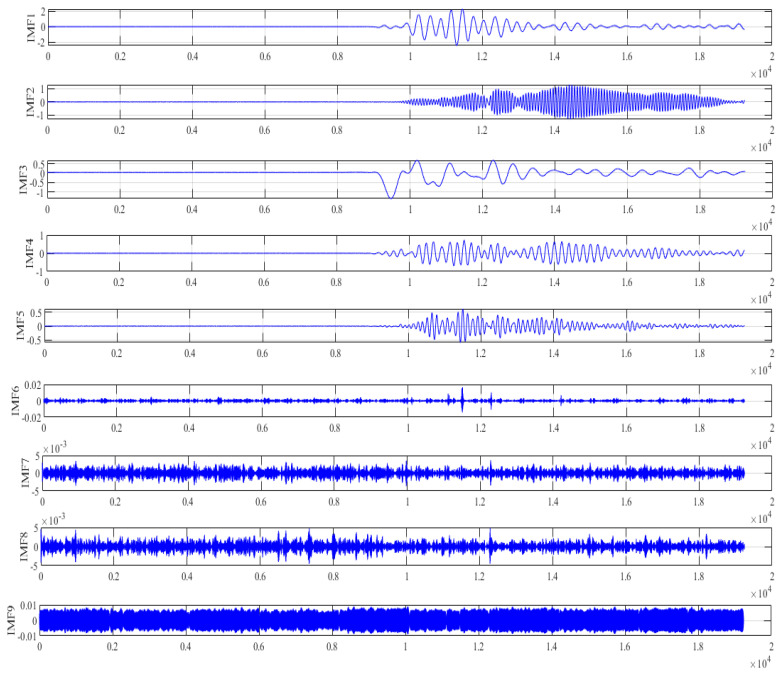
VMD decomposition of the output signal.

**Figure 15 micromachines-13-00891-f015:**
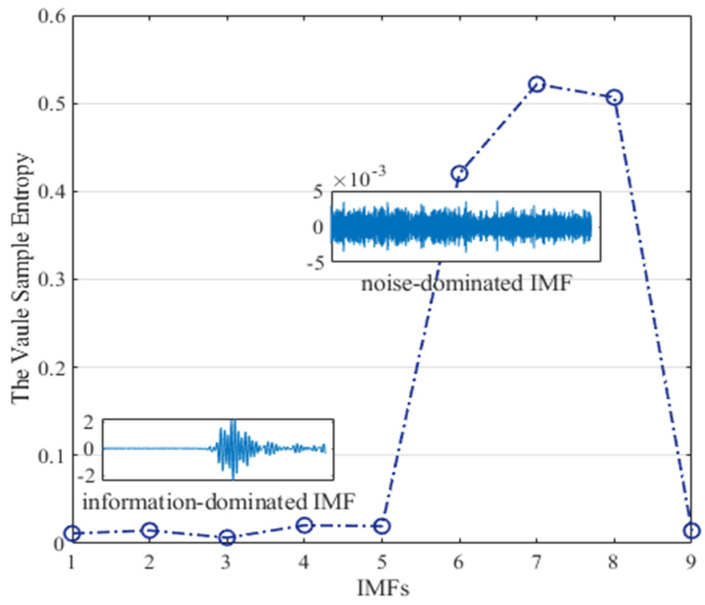
The classification of the IMFs.

**Figure 16 micromachines-13-00891-f016:**
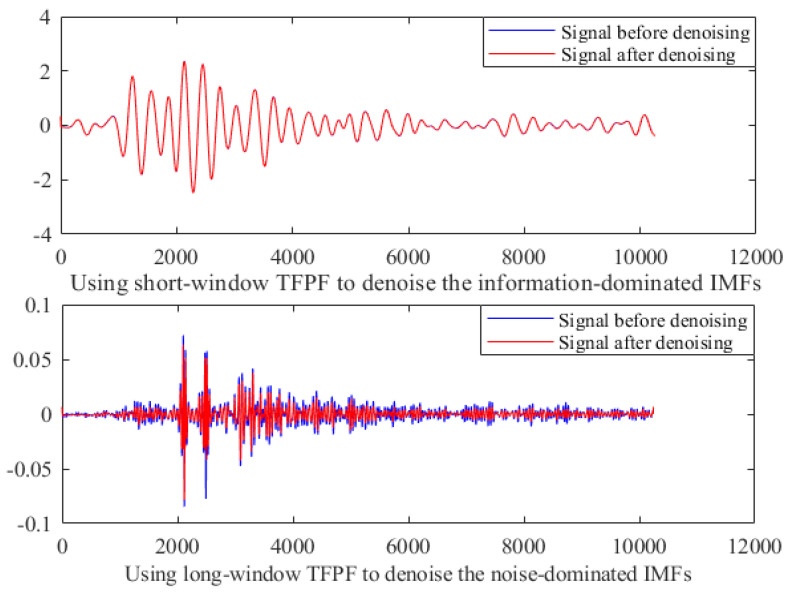
TFPF with different window lengths is selected for denoising.

**Figure 17 micromachines-13-00891-f017:**
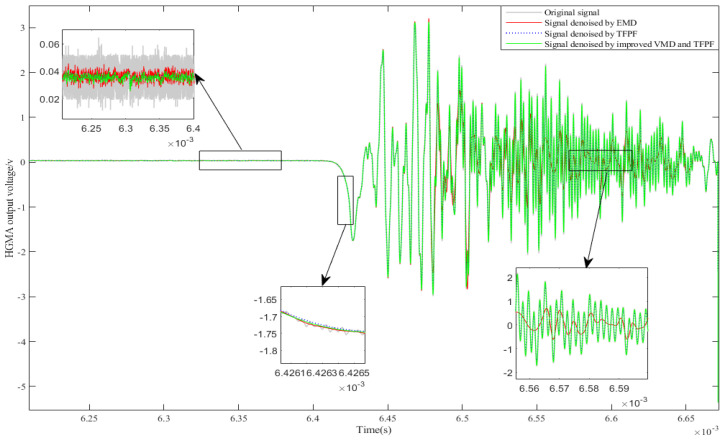
The denoising results of HGMA output signal.

**Figure 18 micromachines-13-00891-f018:**
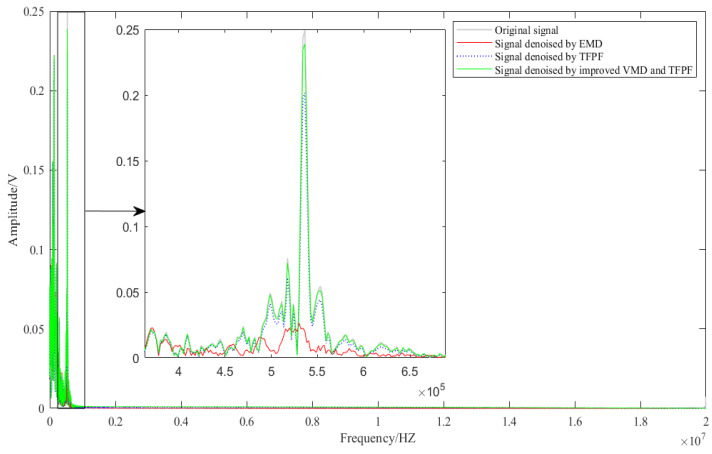
The spectrum of different denoising signals.

**Table 1 micromachines-13-00891-t001:** The specific values of HGMA’s structural parameters.

	HGMA’s Beam			HGMA’s Mass		
**Structural parameters**	length (*a*_1_)	width (*b*_1_)	height (*c*_1_)	length (*a*_2_)	width (*b*_2_)	height (*c*_1_)
**Vaule/μm**	350	800	80	800	800	200

**Table 2 micromachines-13-00891-t002:** The parameter list of the constructed signal.

*f* _1_	*f* _2_	*f* _3_	*f* _4_	*f* _5_	*A_m_*	*g*	*T_m_*
55 Hz	25 Hz	30 Hz	235 Hz	500 Hz	1	4	0.1

**Table 3 micromachines-13-00891-t003:** The denoising performance of different denoising methods.

Denoising Method	SNR	RMSE
**Improved VMD and TFPF**	20.987	0.0603
**EMD**	4.6465	0.3435
**TFPF**	18.9635	0.0726

## Data Availability

Not applicable.

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
