# Peer review of "A Hybrid Algorithm for Noise Suppression of MEMS Accelerometer Based on the Improved VMD and TFPF"

_micromachines, 2022, doi:10.3390/mi13060891_

Round 1

Reviewer 1 Report

In this article, the improved VMD and TFPF are proposed to denoise the HGMA’s output signal. The experimental results show that the improved VMD and TFPF denoising method has the smaller signal distortion, stronger denoising ability, the highest SNR, and lowest RMSE, so it can be adopted to denoise the output signal of the high-g mems accelerometer to improve its accuracy. Before the acceptance, the authors need to answer the following questions.

  1. In part 3, please indicate the source of the HGMA, for example, a reference, a datasheet, or the fabrication process that you performed to fabricate the device.

  1. Please describe the Wheatstone bridge of the HGMA, and please list the fundamental parameters, like sensitivity, resolution, dynamic range, and so on.

Reviewer 2 Report

This paper proposed the improved VMD and TFPF to denoise the HGMA’s output signal. By comparing the denoising results of different denoising algorithms show that the improved VMD and TFPF denoising method can be adopted to denoise the output signal of the high-g mems accelerometer to improve its accuracy. The paper is well organized and well written, but the paper needs minor revision before publication, and I have some other suggestions that may help the paper to be better for readers:

  1. In the fourth paragraph of the third section, the author first explains that the working mode is the first-order mode and the fourth-order mode cannot be excited, so the end of this paragraph is a little redundant, and it is suggested that the author delete it.
  2. The order in which the charts appear in the text should correspond to the text, that is, table 3 should appear after Figure 18.
  3. The English should be further polished, some sentences should be written in a better format.
